# Arrangement of Hydrogen Bonds in Aqueous Solutions of Different Globular Proteins

**DOI:** 10.3390/ijms231911381

**Published:** 2022-09-27

**Authors:** Amber R. Titus, Pedro P. Madeira, Luisa A. Ferreira, Alexander I. Belgovskiy, Elizabeth K. Mann, Jay Adin Mann, William V. Meyer, Anthony E. Smart, Vladimir N. Uversky, Boris Y. Zaslavsky

**Affiliations:** 1Cleveland Diagnostics, 3615 Superior Ave., Cleveland, OH 44114, USA; 2Centro de Investigacao em Materiais Ceramicos e Compositos, Department of Chemistry, 3810-193 Aveiro, Portugal; 3Department of Physics, Kent State University, Kent, OH 44242, USA; 4Department of Chemical and Biomolecular Engineering, Case Western Reserve University, Cleveland, OH 44242, USA; 5Scattering Solutions, Inc., Cleveland, OH 44242, USA; 6Scattering Solutions, Inc., Costa Mesa, CA 02138, USA; 7Department of Molecular Medicine and Byrd Alzheimer’s Research Institute, Morsani College of Medicine, University of South Florida, 12901 Bruce B. Downs Blvd., Tampa, FL 33612, USA

**Keywords:** Fourier transform infrared spectroscopy, water structure, protein solution, hydrogen bond

## Abstract

This work presents the first evidence that dissolved globular proteins change the arrangement of hydrogen bonds in water, with different proteins showing quantitatively different effects. Using ATR-FTIR (attenuated total reflection—Fourier transform infrared) spectroscopic analysis of OH-stretch bands, we obtain quantitative estimates of the relative amounts of the previously reported four subpopulations of water structures coexisting in a variety of aqueous solutions. Where solvatochromic dyes can measure the properties of solutions of non-ionic polymers, the results correlate well with ATR-FTIR measurements. In protein solutions to which solvatochromic dye probes cannot be applied, NMR (nuclear magnetic resonance) spectroscopy was used for the first time to estimate the hydrogen bond donor acidity of water. We found strong correlations between the solvent acidity and arrangement of hydrogen bonds in aqueous solutions for several globular proteins. Even quite similar proteins are found to change water properties in dramatically different ways.

## 1. Introduction

All biological processes, specifically protein folding, biomolecular recognition, hydrophobic effects, protein-ligand binding, and liquid-liquid phase separation in vivo only occur with the active participation of water [1,2,3]. The role of water in many of these biological processes has been mostly ignored [4]. Recently, the active role of water in biological processes has become a subject of focused studies [5,6,7,8]. For example, human heat shock protein HSP6 [9] and dehydrins (dehydration proteins) of various molecular weights [10] alter the solvent features of water. These solvent features in solutions were characterized with solvatochromic dyes [9,10] and included the solvent dipolarity/polarizability (π*) representing dipole-dipole and dipole-induced dipole interactions, solvent hydrogen bond donor acidity (α), and solvent hydrogen bond acceptor basicity (β). We established [11] that one or two of these properties strongly correlate with various physicochemical properties of aqueous solutions of various solutes, such as activity coefficient, osmotic pressure, viscosity, surface tension, and relative permittivity. The effects of nonionic polymers on the solvent features of water are important in macromolecular crowding [12,13]. The excluded volume introduced by macromolecular crowding was earlier thought to dominate the functional and structural properties of proteins and nucleic acids both in vivo and in model conditions in vitro. Phenomena such as folding mechanisms, conformational stability, aggregation propensity, and interactions with other molecules earlier ascribed to space restrictions are now shown to be dependent upon the structure of the water in which they are dissolved. Nonionic polymers are typically used as model crowders, but analysis of their influence on the solvent features of aqueous media [12,13] indicates that they act by changing the solvent properties of water.

In systems showing phase separation, the solvent properties of water differ between phases [14]. These differences govern partitioning between two phases of various solutes ranging from small organic compounds to proteins [15] and are strongly correlated with the interfacial tension between the phases [16].

Solvatochromic dyes permit the estimation of changes in the properties of water in various solutions [11,12,13,14,15] but may be used only for proteins that do not bind to aromatic compounds. This method may not therefore be used for many proteins such as serum albumin or lysozyme. We recently used ATR-FTIR spectroscopy to show that the solvent features of water are strongly correlated with hydrogen bond arrangement in solutions of various compounds [17,18,19,20,21,22]; here we use it to explore the rearrangement of hydrogen bonds in aqueous solutions of several globular proteins.

## 2. Results and Discussion

Figure 1 shows an example of decomposition of the OH-stretch band into four different Gaussian components in a protein solution. Analogous data for the 0.15 M NaCl in 0.01 M Na-phosphate buffer, pH 7.4 (PBS) and human serum albumin at a concentration of 100 mg/mL in PBS are presented in Appendix A. The ratio of the water subpopulations/clusters changes differently in the presence of different proteins.

Estimates of the relative contributions of the Gaussian components I (3080 cm^−1^), component II (3230 cm^−1^), component III (3400 cm^−1^), and component IV (3550 cm^−1^) for all the proteins examined here at various concentrations are listed in Appendix A. Concentration dependences of the relative contributions of the Gaussian components I-IV for human and bovine albumins, ovalbumin, and human γ-globulin are plotted in Figure 2a–d. Analogous dependences for lysozyme, β-lactoglobulins A and B, and trypsinogen are shown in Figure 3a–d. We display the eight proteins, measured separately, in two groups for clarity.

The contribution of Gaussian component I (3080 cm^−1^), assigned to water molecules with four tetrahedrally-arranged hydrogen bond subpopulations, increases linearly with concentration for all globular proteins explored here, whereas that for Gaussian component II (3230 cm^−1^), assigned to water molecules with four distorted hydrogen bonds, decreases linearly with the same materials and constraints. There is no systematic apparent relationship between the slope of Gaussian component I and protein molecular weight.

For each examined protein, the fraction of ice-like water structure (component I) and the fraction of distorted ice-like water structure (component II) both change linearly with increasing protein concentration, but in opposite directions (Figure 2a,b and Figure 3a,b). The relative change of component III increases linearly only for γ-globulin. Component IV decreases linearly for bovine serum albumin, ovalbumin, and γ-globulin, and becomes non-linear for human serum albumin, lysozyme, β-lactoglobulins A and B, and trypsinogen. This suggests that the effects of different proteins on the structure of water depend strongly on the nature and spatial arrangement of the solvent-exposed protein groups.

The rather unexpected finding is how significantly different are the effects of two β-lactoglobulins A and B on hydrogen bond arrangement in aqueous solution (Figure 3a,c). These two proteins differ at only two positions: Gly64 in β-lactoglobulin B substitutes for Asp64 in β-lactoglobulin A, and Ala118 in β-lactoglobulin B substitutes for Vall118 in β-lactoglobulin A. Both proteins form dimers at pH 7.4. All the observed differences in the effects of various proteins on rearrangements of hydrogen bonds in aqueous solutions may be due to differences in the nature and steric arrangements of the solvent-exposed residues in these proteins.

For the proteins examined here, the highest concentrations of 200–300 mg/mL correspond to crowded solutions [23,24,25,26]. In our previous studies [12,13], we found that the crowding effects may be described in terms of crowder-induced changes in the solvent properties of aqueous media. We were unable to find any numerical information in the literature about crowding effects of proteins. Analysis of the possible role of these proteins as crowding agents is outside of the scope of this study.

Analysis of the data in Appendix A shows a strong linear correlation between the α_24_ parameter and the relative contributions of Gaussian components II (3230 cm^−1^) and III (3400 cm^−1^) for a given protein. The correlations observed are presented graphically in Figure 4a–d. For (a) and (b) we observe dependence only on the value of Gaussian component III, whereas for (c) and (d) the dependence is upon both Gaussian components II and III.

As above, solvatochromic dyes typically interact with most proteins and therefore cannot be used to estimate the solvent properties of water in protein solutions. In aqueous solutions of ionic liquids, however, hydrogen bond donor acidity, α_24_, was evaluated with the pyridine-N-oxide probe using the ^13^C NMR chemical shift [27]. Here we used this NMR technique for the first time to evaluate α_24_ in solutions of globular proteins with the results presented in Figure 4 and Appendix A.

These data show that the relative contributions of Gaussian components II (3230 cm^−1^) and/or III (3400 cm^−1^) are strongly correlated with solvent hydrogen bond donor acidity in protein solutions. Gaussian components II and III change much more significantly with protein concentrations than components I and IV. This implies that the changes in the fractions of subpopulations of water with four distorted hydrogen bonds and with four and three hydrogen bonds determine the solvent hydrogen bond acidity of water. Similar relationships were previously reported [20] for solutions of nonionic polymers, inorganic salts, and several small organic compounds. Data presented in Figure 4 show that changes in the arrangement of hydrogen bonds of water in protein solutions cause changes in the solvent properties of water.

Many publications study the effects of water on the properties of proteins [28,29,30,31,32,33,34]. However, most of these studies explore how aqueous media affect protein folding and recognition via hydrophobic effects, and how osmolytes and denaturants alter protein properties. These studies do not consider the possibility that proteins themselves may change the properties of water. They also do not consider the possibility that such solute-changed water can, in its turn, affect solutes differently. According to P. Ball “Despite this status, water’s roles in sustaining life are still imperfectly understood and have until the past several decades been routinely underestimated … The common picture was that of a passive matrix: a solvent that simply acts as a vehicle for the diffusive motions of functional biological macromolecules, such as proteins and nucleic acids. It is now clear that, on the contrary, water plays an active role in the life of the cell over many scales of time and distance …” [2]. The data presented here provide additional support for our hypothesis [11] that solute-induced rearrangement of hydrogen bonds in aqueous solutions can lead to changes in the solvent properties of water, which therefore may describe the water activity and other physicochemical properties of solutions.

## 3. Materials and Methods

### 3.1. Materials

The following specimens were purchased from Sigma-Aldrich (St. Louis, MO, USA): (1) human serum albumin (fatty acid and human globulin free (~99%)), (2) albumin from bovine serum, (3) human γ-globulin, (4) β-lactoglobulin A from bovine milk (>90%), (5) β-lactoglobulin B from bovine milk (>90%), (6) lysozyme from chicken egg white, (7) trypsinogen from bovine pancreas, (8) ovalbumin, and (9) pyridine-N-oxide (PyO) with a purity of 95% for the NMR measurements and analysis.

### 3.2. Methods

#### 3.2.1. ATR-FTIR Measurements

ATR-FTIR spectra for each sample were measured in two separately prepared solutions using an Alpha II FT-IR spectrometer equipped with Platinum single reflection ATR single reflection diamond ATR module (Bruker Scientific, LLC, Billerica, MA, USA). All measurements were performed at ambient temperature (approximately 23 °C) using 24 scans for each sample and 24 scans for background in the spectral range of 4000–1000 cm^−1^ with resolution of 4 cm^−1^. The spectra were reproducible to better than 1 cm^−1^.

#### 3.2.2. Analysis of ATR-FTIR Spectra

ATR-FTIR spectra were analyzed using custom software written in Wolfram Mathematica and run under version 12. The software analyzed the OH-stretch band by fitting the data using the ‘NonlinearModelFit’ function, with a model using the sum of three, four, or five Gaussian distributions with floated peak frequencies, amplitudes, and widths. Using this model, fitting a baseline is necessary where real values may extend beyond the available measured dataset or the data may have a residual offset. We found that the best and most reliable fits gave peak frequencies within the error band of values from literature [22], which were therefore fixed for all later fitting routines. These values, which we confirmed earlier [17,20], are 3080 cm^−1^, 3230 cm^−1^, 3400 cm^−1^, and 3550 cm^−1^, represented by $\mu_{1-4}$ respectively. The fit includes the four Gaussian expressions identified by the subscript, with baseline B and normalization by A to the measured total amplitude, where $\sigma_{i}$ is the standard deviation of each Gaussian component, and $a_{i}$ is the amplitude contribution of each:$$B+A\overset{4}{\underset{i=1}{\sum }}\frac{a_{i}}{2\sigma_{i}\surd \pi }e^{-\left[\frac{1}{2}{\left(\frac{x-\mu_{i}}{\sigma_{i}}\right)}^{2}\right]}.$$

The program displays graphically the raw data, model function fit, and individual Gaussian distributions. Also available are all fitted parameter values together with a suite of boundary metrics quantifying confidence for each fit.

In FTIR spectra, vibrational bands such as the C–O–H stretch mode of alcohols and biomolecules together with their N-H stretch (Amide A) modes overlap the broad OH-stretch band of water. We suggest that this OH-stretch band in solutions of the proteins examined here represents water sufficiently closely to support the results presented in this paper. This is justified by two observations. First, the molar concentration of water exceeds those of the protein functional groups by several orders of magnitude, even at the highest concentrations used in this study. Second, in the region of interest (2600–3800 cm^−1^), water absorbs more than ten times more strongly than any protein functional group.

ATR-FTIR spectroscopic analysis of the OH-stretch band showed that the hydrogen bonding of water depends on specific solutes, such as inorganic salts, trimethylamine N-oxide, urea, the polymers PEG, PVP, and a copolymer of ethylene glycol and propylene glycol (Ucon), and their concentrations [20]. For every compound examined, the minimal unstructured residuals were obtained by fitting the measured OH-stretch band with four Gaussian components peaking at 3080, 3230, 3400, and 3550 cm^−1^. We confirmed the quality of this fit by calculating the correlation matrix, assuring the independence of the variables.

Our results show that this simple model of exactly four Gaussian components suggests the simultaneous coexistence of four subpopulations of water with different H-bond arrangements; water with four tetrahedrally arranged hydrogen bonds (3080 cm^−1^), water with four distorted hydrogen bonds (3230 cm^−1^), water with loosely arranged three or four hydrogen bonds (3400 cm^−1^), and water with three, two, or one hydrogen bond(s) (3550 cm^−1^) [20]. The proportion of each subpopulation depends on the properties and concentration of the solute. However, the physical distribution, geometry, structure, molecular arrangement, and scale of each subpopulation are currently unknown. Previously reported solvent properties of water [20], such as solvent dipolarity/polarizability, π*, solvent H-bond donor acidity, α, and solvent H-bond acceptor basicity, β, correlate strongly with the fractional contributions of subpopulations of water.

This approach was used successfully to analyze the coexisting phases of several aqueous two-phase systems [17]. We apply the same model here for the analysis of water in solutions of different globular proteins.

#### 3.2.3. NMR Measurements

NMR chemical shifts (d) in ppm were determined using a Bruker Avance 300 spectrometer (operating at 300.13 MHz for ^1^H and 75.47 MHz for ^13^C NMR). Solutions containing 0.25 mol dm^−3^ of pyridine-N-oxide in each sample aimed to be characterized, and a solution of tetramethylsilane (TMS) in pure deuterated water (99.9% D) as an internal standard, were used in NMR tubes adapted with coaxial inserts. The TMS/D_2_O solution was always used as the inner part of the concentric tubes, while each sample was used in the outer part of the NMR tube. Using this approach, it is possible to guarantee that the TMS standard and D_2_O are not in direct contact with the sample, avoiding possible interferences or deviations in the ^13^C NMR chemical shifts. The Mnova software (Santiago de Compostela, Spain) was used for data processing. The parameter α_24_ is determined as the ^13^C NMR chemical shift of carbon 2, relative to that of carbon 4 of pyridine-*N*-oxide (PyO) probe and represents the solvent hydrogen bond donor acidity of aqueous media [27] in protein solution.

## 4. Conclusions

The earlier model of four different structures for water represented by exactly four Gaussian spectral components and describing the OH-stretch band in the ATR-FTIR spectra of solutions of non-ionic polymers and inorganic salts [20], is also applicable to protein solutions.The effects of different proteins on the structure of water depend strongly on the nature and spatial arrangement of the solvent-exposed protein groups.The solvent hydrogen bond donor acidity in aqueous protein solutions (α_24_) may be estimated by the NMR technique described in [27], which has not previously been applied to protein solutions.For the globular proteins in the aqueous solutions examined here, the contributions of Gaussian components II and III are strongly correlated with the hydrogen bond donor acidity of water.

## Figures and Tables

**Figure 1 ijms-23-11381-f001:**
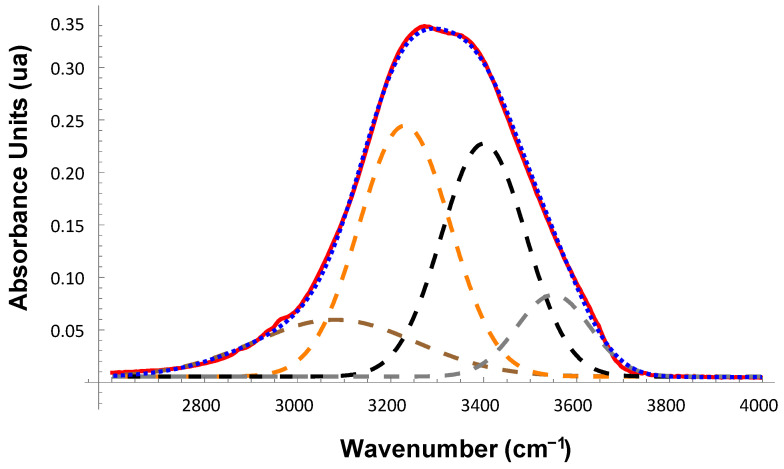
Examples of the ATR-FTIR spectra of OH-stretch band in the solvent for 250 mg/mL β-lactoglobulin A in 0.15 M NaCl in 0.01 M Na-phosphate buffer, pH 7.4 (PBS). The blue dotted line is the measured absorption spectrum, and the red envelope is the best fit of the sum of our four Gaussian components (dashed lines at positions (3080 cm^−1^ (dark orange), 3230 cm^−1^ (orange), 3400 cm^−1^ (black), and 3550 cm^−1^ (gray)). Experimental data and fit are visually almost indistinguishable.

**Figure 2 ijms-23-11381-f002:**
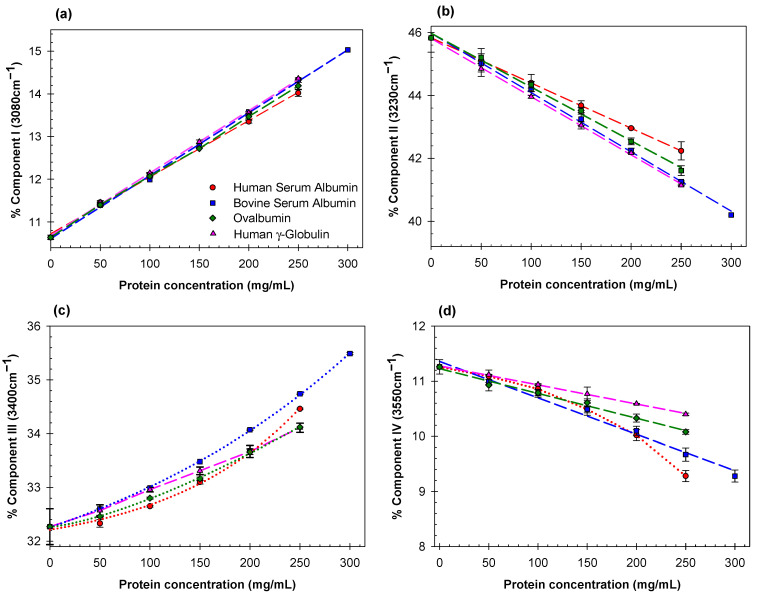
Concentration dependence of the relative contributions of the Gaussian components (**a**) I, (**b**) II, (**c**) III, and (**d**) IV for human serum albumin (red circles), bovine serum albumin (blue squares), ovalbumin (green diamonds), and human γ-globulin (magenta triangles).

**Figure 3 ijms-23-11381-f003:**
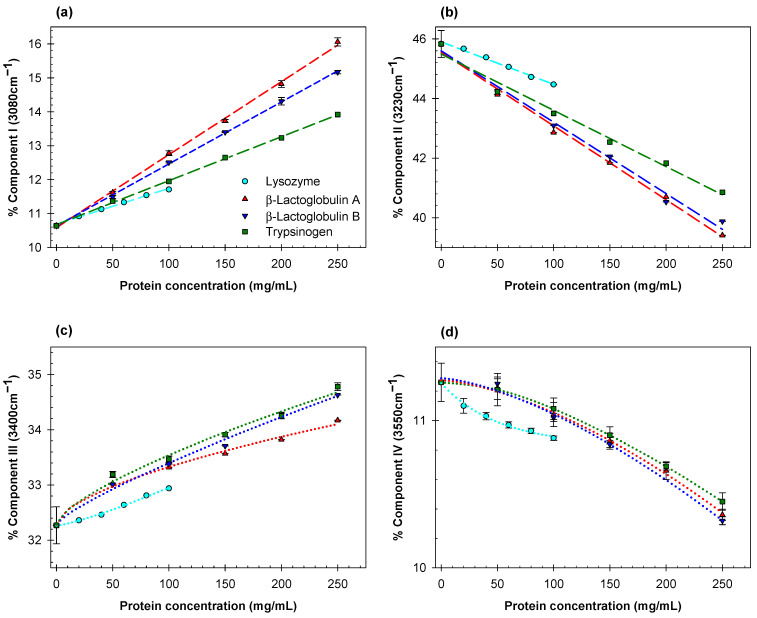
Concentration dependences of the relative contributions of the Gaussian components I (**a**), II (**b**), III (**c**), and IV (**d**) for chicken egg lysozyme (cyan circles), β-lactoglobulins A (red triangles) and B (blue inverted triangles), and trypsinogen (green squares).

**Figure 4 ijms-23-11381-f004:**
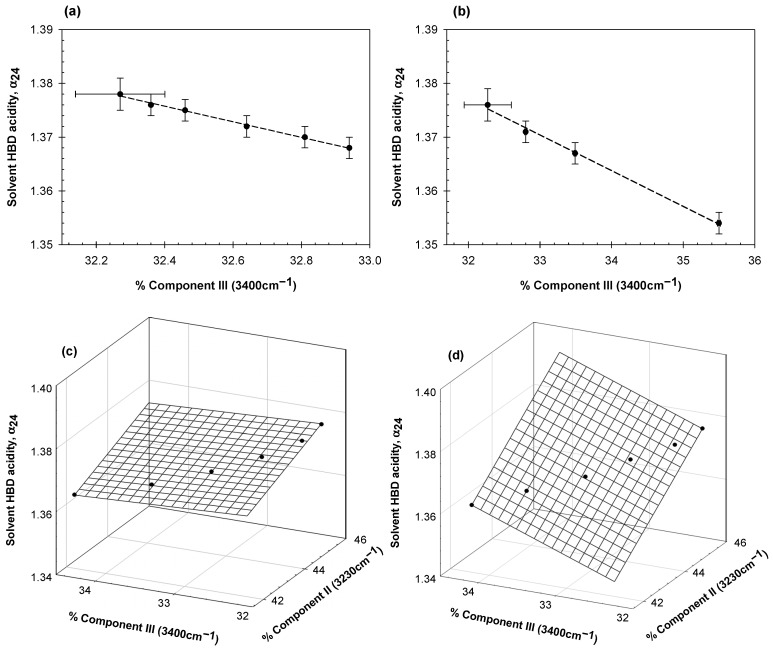
Relationship between the solvent hydrogen bond donor acidity, α_24_, and the relative contributions of Gaussian component III for (**a**) lysozyme and (**b**) bovine serum albumin, and the relative contributions of Gaussian components II and III for (**c**) human serum albumin and (**d**) ovalbumin.

## Data Availability

All raw ATR-FTIR data along with the Wolfram Mathematica code used are available upon request.

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
