# Peer review of "Arrangement of Hydrogen Bonds in Aqueous Solutions of Different Globular Proteins"

_ijms, 2022, doi:10.3390/ijms231911381_

Round 1

Reviewer 1 Report

Authors Amber R. Titus, et.al. in their manuscript entitled "Arrangement of hydrogen bonds in aqueous solutions of different globular proteins" presented as they stated the first experimental evidence that solubilized proteins alter the arrangement of hydrogen bonds in water by analyzing the OH stretching band in ATR spectra of different protein solutions. Based on their previous studies, they decomposed the OH stretching into four components associated with different structures of water molecules; from ice-like to the IV component at 3550 cm-1.

Unfortunately, I cannot agree with all the above claims.

1. This paper is certainly not the first experimental attempt to show whether proteins change water structure

2. Although they have already published several articles using the decomposition of the OH water band, it is very likely that this way of interpreting the OH band structure is incorrect.

It is correct that the band shape of the OH stretching reflects the distribution of hydrogen bonds in the water solution. However, these different populations of hydrogen bonds in relation to the geometry and (or) strength are not the only mechanisms that affect the band shape.

It is well known that intra- and intermolecular coupling, as well as interaction with the deformation overtone, significantly perturb the band shape of OH stretching. Therefore, any change in the coupling constants will affect the band shape, which may not be directly related to the structure of the hydrogen bonding network. By analyzing the OH stretching band, one generally does not learn what reasons affected the OH stretching band.

Of course, there is a solution how to eliminate the coupling contributions. Instead of using pure H2O as a solvent, one should use low concentration H2O in D2O (or vice versa). At a low concentration of H2O in D2O, only HDO molecules can be observed, and thus the conditions for intra- and intermolecular coupling are disabled, resulting in a much more symmetrical band shape of the OH stretching of HDO species. By analyzing the HOD stretching we can indeed retrieve the changes in the solvent structure especially when the difference spectroscopy is applied (i.e. by subtracting the spectrum of bulk solvent).

And just a little question about the assignment. How can you assign the band at 3080 cm-1 to ice-like molecules when it is known that the peak position of OH stretching of hexagonal ice is at 3260 cm-1? Everyone would expect the ice-like component at higher wavenumbers as found in the solid state, wouldn't they?

The improper interpretation of the experimental data, the questionable conclusions, and the lack of originality, unfortunately, force me to reject the submitted work.

Author Response

Authors Amber R. Titus, et.al. in their manuscript entitled "Arrangement of hydrogen bonds in aqueous solutions of different globular proteins" presented as they stated the first experimental evidence that solubilized proteins alter the arrangement of hydrogen bonds in water by analyzing the OH stretching band in ATR spectra of different protein solutions. Based on their previous studies, they decomposed the OH stretching into four components associated with different structures of water molecules; from ice-like to the IV component at 3550 cm-1.

Unfortunately, I cannot agree with all the above claims.

  1. This paper is certainly not the first experimental attempt to show whether proteins change water structure

Response.  We are not aware of any published experimental studies of the effects of proteins on the bulk solvent properties of water.  We would be grateful if the reviewer would provide any such references.

  1. Although they have already published several articles using the decomposition of the OH water band, it is very likely that this way of interpreting the OH band structure is incorrect.

It is correct that the band shape of the OH stretching reflects the distribution of hydrogen bonds in the water solution. However, these different populations of hydrogen bonds in relation to the geometry and (or) strength are not the only mechanisms that affect the band shape.

It is well known that intra- and intermolecular coupling, as well as interaction with the deformation overtone, significantly perturb the band shape of OH stretching. Therefore, any change in the coupling constants will affect the band shape, which may not be directly related to the structure of the hydrogen bonding network. By analyzing the OH stretching band, one generally does not learn what reasons affected the OH stretching band.

Of course, there is a solution how to eliminate the coupling contributions. Instead of using pure H2O as a solvent, one should use low concentration H2O in D2O (or vice versa). At a low concentration of H2O in D2O, only HDO molecules can be observed, and thus the conditions for intra- and intermolecular coupling are disabled, resulting in a much more symmetrical band shape of the OH stretching of HDO species. By analyzing the HOD stretching we can indeed retrieve the changes in the solvent structure especially when the difference spectroscopy is applied (i.e. by subtracting the spectrum of bulk solvent).

Response.  We examined these issues in the way suggested by the reviewer [J. Pavelec, D. DiGuiseppi, B.Y. Zaslavsky, V.N. Uversky, R. Schweitzer-Stenner, Perturbation of water structure by water-polymer interactions probed by FTIR and polarized Raman spectroscopy, J. Mol. Liquids, 275(2019) 463-473.].  We showed that coupling effects on perturbation of water structure by two nonionic polymers in their aqueous solutions are negligible.

We emphasize that the overall goal of our work was to show that globular proteins produce noticeable effects on the structure of the water in which they are dissolved.  This is evidenced by the changes that these proteins induce in the ATR-FTIR spectra of the OH-stretch bands of water.

We appreciate the reviewer’s suggestion that the changes in the band shape may not be directly related to the structure of the hydrogen bonding network, however, in our previous studies (referenced above) we showed that the ATR-FTIR spectral changes correlate strongly with changes in solvent properties of polymer solutions as measured by the solvatochromic dyes.  In the current manuscript, we describe our use of NMR to measure the hydrogen bond donor acidity of water and report its strong correlation with the ATR-FTIR data.  Therefore, we can conclude that using ATR-FTIR measurements, with the peak assignments described in the literature, the recorded spectral transformations reflect changes in the relative amounts of the four subpopulations of water structures coexisting in the examined aqueous solutions.

And just a little question about the assignment. How can you assign the band at 3080 cm-1 to ice-like molecules when it is known that the peak position of OH stretching of hexagonal ice is at 3260 cm-1? Everyone would expect the ice-like component at higher wavenumbers as found in the solid state, wouldn't they?

Response.  For the work described in this manuscript, we used well-documented peak assignments from the literature.  Our assignment of sub-bands to various water structures reported in the literature was discussed in detail in the previous publication [N. R. da Silva, L. A. Ferreira, A. I. Belgovskiy, P. P. Madeira, J. A. Teixeira, E. K. Mann, J. A. Mann Jr., W. V. Meyer, A. E. Smart, V. Y. Chernyak, V N. Uversky, B. Y. Zaslavsky, Effects of different solutes on the physical chemical properties of aqueous solutions via rearrangement of hydrogen bonds in water, J. Mol. Liquids, 2021].  Assignment of peak frequencies is based on an oversimplified model used for the decomposition of the OH-stretch band.  The strength of this model, however, is its proven applicability to aqueous solutions of various solutes from inorganic salts, small organic compounds, nonionic polymers to proteins.  Experimentally it provides information about the water structure (arrangement of hydrogen bonds), which is strongly correlated with the solvent properties of water in the aqueous solutions.

The improper interpretation of the experimental data, the questionable conclusions, and the lack of originality, unfortunately, force me to reject the submitted work.

Response.  Although more comprehensive methods for interpreting ATR-FTIR spectral changes may be imagined in the future, the present conclusions are in no way either ‘improper’ or ‘questionable’.

Reviewer 2 Report

The authors have performed ATR-FTIR measurements on 8 different proteins in aqueous solutions to show that the weight of the four Gaussian components of the water spectrum in the 2800-3800 cm-1 region changes depending on the protein concentration. This is surely an interesting result, but I think the authors have to perform a significant revision of the manuscript.

First, it is not correct to write: “The role of water in many of these biological processes has been mostly ignored.” Kauzmann in his famous review of 1959, Adv.Protein Chem. 14 (1959) 1, pointed out the fundamental role played by water for the occurrence of the hydrophobic effect, that is the main actor in protein folding and molecular recognition. See also the review by Ken Dill in Biochemistry 29 (1990) 7133.

Second, the changes are in any case small considering that the protein concentration is increased a lot; the final concentrations should correspond to crowded solutions. This should be underscored in the manuscript. Moreover, there should be articles by the group of Kim Sharp investigating the response of water structure to the addition of different small and large solutes, Acc.Chem.Res. 43 (2010) 231.

Third, I think the authors have to pay attention to the enthalpy-entropy compensation phenomenon that is ubiquitous in water and aqueous solutions; see Dunitz, Chem.Biol. 2 (1995) 709; Sharp, Protein Sci. 10 (2001) 661. The observation of small changes in the 3D hydrogen bonded network of liquid water has not transparent consequences, especially from the thermodynamic point of view.

Fourth, the authors show in Figure 4 the occurrence of a correlation between the solvent hydrogen bond donor acidity and the Gaussian components II and III. Does this imply the absence of a correlation for the Gaussian components I and IV? What is the explanation? Some sentences would be useful for interested readers.

Author Response

The authors have performed ATR-FTIR measurements on 8 different proteins in aqueous solutions to show that the weight of the four Gaussian components of the water spectrum in the 2800-3800 cm-1 region changes depending on the protein concentration. This is surely an interesting result, but I think the authors have to perform a significant revision of the manuscript.

First, it is not correct to write: “The role of water in many of these biological processes has been mostly ignored.” Kauzmann in his famous review of 1959, Adv.Protein Chem. 14 (1959) 1, pointed out the fundamental role played by water for the occurrence of the hydrophobic effect, that is the main actor in protein folding and molecular recognition. See also the review by Ken Dill in Biochemistry 29 (1990) 7133.

Response:  We agree with the reviewer that there is a number of publications from several groups studying the effects of water on properties of proteins.  However, most of these studies have a rather unidirectional view (e.g., water affects protein folding and recognition via the hydrophobic effects) and mostly ignore the effects of solutes on properties of water.  They also do not consider the possibility that such solute-changed water can, in its turn, affect solutes differently.  This is in addition to considerations described by P.Ball: “Despite this status, water’s roles in sustaining life are still imperfectly understood and have until the past several decades been routinely underestimated (2). The common picture was that of a passive matrix: a solvent that simply acts as a vehicle for the diffusive motions of functional biological macromolecules, such as proteins and nucleic acids. It is now clear that, on the contrary, water plays an active role in the life of the cell over many scales of time and distance (3). [P. Ball, Ball, P. (2017). Water is an active matrix of life for cell and molecular biology. Proceedings of the National Academy of Sciences114(51), 13327-13335.]  The corresponding discussion is added to the revised manuscript (lines 150-164, p. 5-6).

Second, the changes are in any case small considering that the protein concentration is increased a lot; the final concentrations should correspond to crowded solutions. This should be underscored in the manuscript. Moreover, there should be articles by the group of Kim Sharp investigating the response of water structure to the addition of different small and large solutes, Acc.Chem.Res. 43 (2010) 231.

Response: We agree with the reviewer’s comment.  In previous studies, we found that crowding effects can be described in terms of the crowder-induced changes in the solvent properties of aqueous media [Ferreira, L.A., Madeira, P.P., Breydo, L., Reichardt, C., Uversky, V.N., Zaslavsky, B.Y. Role of Solvent Properties of Aqueous Media in Macromolecular Crowding Effects. J. Biomol. Struct. Dynam., 34 (1) (2016) 92-103.; L.A. Ferreira, V.N. Uversky, B.Y. Zaslavsky, Role of solvent properties of water in crowding effects induced by macromolecular agents and osmolytes, Mol. BioSystems, 2017, 13, 2551-2563].  We were unable to find any numerical information about crowding effects of proteins in the literature.  The corresponding discussion is added to the revised manuscript (lines 117-122, p. 4).

Third, I think the authors have to pay attention to the enthalpy-entropy compensation phenomenon that is ubiquitous in water and aqueous solutions; see Dunitz, Chem.Biol. 2 (1995) 709; Sharp, Protein Sci. 10 (2001) 661. The observation of small changes in the 3D hydrogen bonded network of liquid water has not transparent consequences, especially from the thermodynamic point of view.

Response:  We agree with the reviewer’s comment that “The observation of small changes in the 3D hydrogen bonded network of liquid water has not transparent consequences, especially from the thermodynamic point of view”.  However, establishing a justifiable relationship between the thermodynamic approach via statistical mechanics and the complex and as yet unspecified spatial and temporal structure of hydrogen bonding is beyond the scope of this paper.  That is why we did not discuss the data obtained from the thermodynamic viewpoint.

We did establish previously that the solvent properties of water in aqueous solutions of different solutes may describe the water activity and other physicochemical properties of solutions [L.A. Ferreira, J. A. Loureiro, J. Gomes, V. N. Uversky, P. P. Madeira, B. Y. Zaslavsky, Why physicochemical properties of aqueous solutions of various compounds are linearly interrelated, J. Mol. Liq., 221 (2016) 116-123.].  The corresponding explanation is added to the revised manuscript (lines 161-164, p. 6).

Fourth, the authors show in Figure 4 the occurrence of a correlation between the solvent hydrogen bond donor acidity and the Gaussian components II and III. Does this imply the absence of a correlation for the Gaussian components I and IV? What is the explanation? Some sentences would be useful for interested readers.

Response: We agree with the reviewer’s comment.  We added an explanation in the manuscript (lines 142-145, p.5).

Round 2

Reviewer 1 Report

Response.  We are not aware of any published experimental studies of the effects of proteins on the bulk solvent properties of water.  We would be grateful if the reviewer would provide any such references.

Answer: J. Am. Chem. Soc. 2017, 139, 3, 1098, Current Opinion in Structural Biology Volume 16, Issue 2, April 2006, Pages 152-159, J. Chem. Phys. 150, 094701 (2019), PNAS 2002, vol. 99, str. 5378-5383, Marechal The Hydrogen Bond and Water Molecules (Elsevier, 2007)…

Response.  We examined these issues in the way suggested by the reviewer [J. Pavelec, D. DiGuiseppi, B.Y. Zaslavsky, V.N. Uversky, R. Schweitzer-Stenner, Perturbation of water structure by water-polymer interactions probed by FTIR and polarized Raman spectroscopy, J. Mol. Liquids, 275(2019) 463-473.].  We showed that coupling effects on perturbation of water structure by two nonionic polymers in their aqueous solutions are negligible.

Answer: Even if you say that the coupling is weak, why is the shape of OH stretching so different from completely uncoupled OH stretching (see, e.g., Bakker and Skinner, Chem. Rev. 2010, 1498)? It is obvious that many unfortunately unknown factors influence the structure of OH stretch.

Response: We emphasize that the overall goal of our work was to show that globular proteins produce noticeable effects on the structure of the water in which they are dissolved.  This is evidenced by the changes that these proteins induce in the ATR-FTIR spectra of the OH-stretch bands of water.

Answer: Why you did not use difference spectroscopy?

We appreciate the reviewer’s suggestion that the changes in the band shape may not be directly related to the structure of the hydrogen bonding network, however, in our previous studies (referenced above) we showed that the ATR-FTIR spectral changes correlate strongly with changes in solvent properties of polymer solutions as measured by the solvatochromic dyes.  In the current manuscript, we describe our use of NMR to measure the hydrogen bond donor acidity of water and report its strong correlation with the ATR-FTIR data.  Therefore, we can conclude that using ATR-FTIR measurements, with the peak assignments described in the literature, the recorded spectral transformations reflect changes in the relative amounts of the four subpopulations of water structures coexisting in the examined aqueous solutions.

Answer: In general, it is very difficult to answer the question about the influence of all possible factors on the shape of the OH stretch band. However, it is obvious that the band structure of dissolved H2O in D2O has a completely different shape. And it is much safer to use this system for water structure analysis (BakkerË›& Skinner Chem Rev 2010. 1498). It can be easily correlated with the populations of hydrogen bonds with different strength and further with the different structure of water molecules.
Moreover, you have included the ATR spectra. So, the correct unit of the y-axis is the ATR absorbance and not the absorbance. The spectrum is not an (almost pure) absorption spectrum as in transmission measurements, but may include contributions from reflections. And these contributions can vary as the composition of the sample changes.

Response.  For the work described in this manuscript, we used well-documented peak assignments from the literature.  Our assignment of sub-bands to various water structures reported in the literature was discussed in detail in the previous publication [N. R. da Silva, L. A. Ferreira, A. I. Belgovskiy, P. P. Madeira, J. A. Teixeira, E. K. Mann, J. A. Mann Jr., W. V. Meyer, A. E. Smart, V. Y. Chernyak, V N. Uversky, B. Y. Zaslavsky, Effects of different solutes on the physical chemical properties of aqueous solutions via rearrangement of hydrogen bonds in water, J. Mol. Liquids, 2021].  Assignment of peak frequencies is based on an oversimplified model used for the decomposition of the OH-stretch band.  The strength of this model, however, is its proven applicability to aqueous solutions of various solutes from inorganic salts, small organic compounds, nonionic polymers to proteins.  Experimentally it provides information about the water structure (arrangement of hydrogen bonds), which is strongly correlated with the solvent properties of water in the aqueous solutions.

Answer: Your proposed assignment is completely different from that of Bakker and Skinner, who showed the different populations of water molecules in terms of the number of established hydrogen bonds.

Response.  Although more comprehensive methods for interpreting ATR-FTIR spectral changes may be imagined in the future, the present conclusions are in no way either ‘improper’ or ‘questionable’.

Answer: These methods already exist. First, calculate the 2*n*k spectrum of dissolved H2O in D2O and analyse the band shape of the OH stretching mode that correlates with different types of water molecules.  You can skip the calculation of the 2*n*k spectrum if you only want to describe changes. Finally, if you are really only interested in the changes in bulk water due to the presence of protein, why did not you just use differential spectroscopy and show these differences in the stretching region? Of course, if you use pure H2O, you still lack the explanation of the causes leading to the changes in the band shape of the OH stretching.

Reviewer 2 Report

I am satisfied with the changes performed by the authors. The revised version of the manuscript is suitable to be published.